# Contact Fatigue Behavior Evolution of 18CrNiMo7-6 Gear Steel Based on Surface Integrity

Luji Wu [1,2], Yongxin Lv [1], Yalong Zhang [3,*], Anhu Li [2] and Vincent Ji [4]

1   Zhengzhou Research Institute of Mechanical Engineering Co., Ltd., Zhengzhou 450052, China
2   School of Mechanical Engineering, Tongji University, Shanghai 200092, China
3   School of Aerospace Engineering, Zhengzhou University of Aeronautics, Zhengzhou 450046, China
4   ICMMO/SP2M, UMR CNRS 8182, Paris-Saclay University, 91405 Paris, France
*   Correspondence: zhangyalong@zua.edu.cn

**Abstract:** In this work, the surface integrity (surface morphology, microstructure, microhardness, residual stress) of contact fatigue (CF) samples with different numbers of running cycles was comprehensively studied. Based on typical working conditions, a fatigue life evaluation method was proposed based on the evolution law of surface integrity. The CF with different numbers of running cycles revealed that the average grain size decreased with the increase in the number of running cycles, and the surface microhardness, residual stress and surface roughness Ra increased first and then decreased. In addition, the relationships between different surface integrity parameters and fatigue life were plotted. Moreover, based on the fatigue life profiles, the running state and remaining life of gear samples can be evaluated.

**Keywords:** contact fatigue behavior; running cycles; surface integrity; failure analysis; 18CrNiMo7-6 gear steel

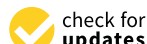



## 1. Introduction

Gears are the most widely used form in transmission systems. Compared with other mechanical transmission forms, gear transmission can change speed, torque and direction, which also has the advantages of high transmission efficiency, accuracy, a stable transmission ratio, and a large power range [1–3]. Due to the high working speed and large load, the gear needs to withstand the shear stress and impact force under the action of circulation. However, gear fatigue is an inevitable phenomenon in the service life. Gear fatigue can eventually lead to serious failures, including surface pitting, spalling and fracture [4,5]. These failures can significantly reduce the remaining service life, and even affect the normal operation of the mechanical equipment. For example, Bejger et al. argued that gears are subjected to alternating loads due to wind speed changes and free braking pulses, which makes them one of the most fragile components of a wind turbine with low reliability, and gear fatigue failure led to the abnormal operation of wind power generation devices [6].

The gear service life is directly related to the performance characteristics of the tooth surface [7,8]. Obtaining the surface integrity characteristics of the gear surface is an important step to improve service life. In order to effectively monitor the degradation of gear material performance during the service cycle, Feng et al. proposed gear remaining service life and fatigue propagation monitoring technology based on vibration [9]. Xiang et al. developed a new type of long short-term memory neural network with weight amplification to predict gear remaining life, which was based on fusing the time-domain and frequency-domain features of vibration signals [10]. Wang et al. used the Archard model to formulate the gear tooth wear and predicted gear service life based on tooth surface wear [11]. Moreover, a variety of surface strengthening techniques have also been applied to gears to improve their fatigue life, such as laser shock peening, surface ultrasonic rolling, pneumatic shot peening, etc. [12–14]. Zhang et al. [15] found that surface ultrasonic rolling

can improve the surface integrity and fatigue life of the 17Cr2Ni2MoVNb gear steel due to the surface work hardening layer. Qu et al. [16] revealed that shot peening can improve the fatigue life of shaft steel due to the formation of dislocation tangles. In addition, some studies have proposed crack initiation and fatigue life prediction based on surface damage and wear performance [17,18].

Although a series of research achievements have been made in improving the surface integrity, fatigue life and service performance of gear samples, the variation law of tooth surface integrity during the service period has not yet been established. Therefore, the aim of this work is to analyze the variation in surface integrity with different numbers of running cycles, which meant testing sample surfaces at different cycle counts. The results provide basic data for providing tooth surface characteristic parameters and later gear maintenance.

## 2. Materials and Methods

The chemical components (wt.%) of 18CrNiMo7-6 steel were C 0.18, Si 0.28, Mn 0.7, Cr 1.6, Ni 1.48, Al 0.097, Mo 0.29 and Fe balance. The heat treatment process was as follows: the carburizing temperature was 920~930 °C for 45 h. After carburizing treatment, the samples were furnace-cooled to 850 °C for 0.5 h and quickly quenched in oil for 0.5 h, and the samples were tempered at 160 °C for 3 h.

CF tests were performed on a roller fatigue tester, and vibration signals were used to monitor contact surface damage. The fatigue sample size and a physical image of the roller fatigue tester are shown in Figure 1. Mobil lubricant (APIGL 80W90) was selected for lubrication, and CF samples of 18CrNiMo7-6 steel were processed according to standard YB/T5345-2014 [19]. A scanning electron microscope (SEM, NOVA NANOSEM 430, Financial Education Initiative, Hillsboro, OR, USA) was used to observe the surface morphology. Electrolytic polishing was used to prepare the target surface for electron backscattering diffraction (EBSD) characterization. The EBSD characterization was carried out on HKL Symmetry equipment (Oxford Instruments, Abingdon, UK), with a 20 kV accelerating voltage and scanning step size of 0.07 μm. A durometer (SCTMC, HV-50, Shangcai Instruments, Shanghai, China) was used to evaluate microhardness, and the applied load and dwelling time were 200 g and 15 s, respectively. The residual stresses were measured by using a X-ray diffraction residual stress tester (Proto LXRD, Michigan, ON, Canada) with the $\sin2\psi$ method and Cr-K$\alpha$ radiation. A roughness tester was used to evaluate surface roughness (Mahr-300C, Mahr Group, Limbach-Oberfrohna, Germany).

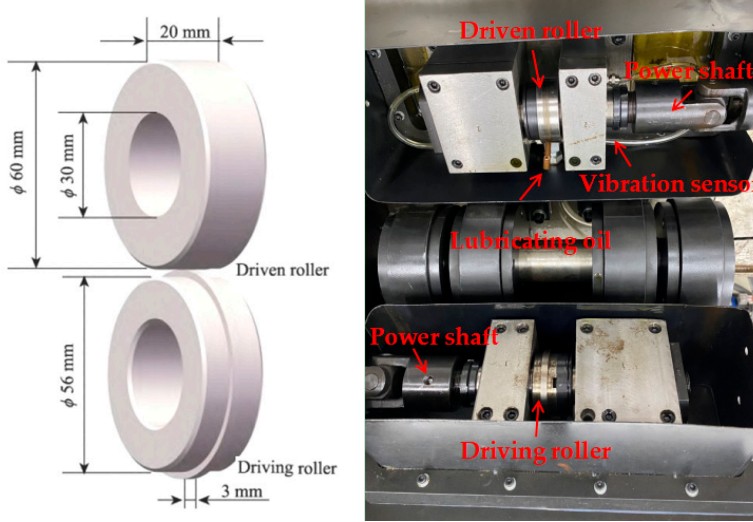

**Figure 1.** Fatigue sample size and testing machine.

The Hertzian stress between the driven roller and driving roller was calculated using the following equation:

$$P_0 = \frac{1}{\pi} \sqrt[3]{6F \times \left( \frac{\frac{1}{R_1} + \frac{1}{R_2}}{\frac{1-v_1^2}{E_1} + \frac{1-v_2^2}{E_2}} \right)^2} \tag{1}$$

where σ is the contact stress; $F$ is the loading force; $R_1$ is the radius of driven roller, $R_2$ is the radius of the driving roller, $E_1$ and $E_2$ are the elastic modulus of the samples, respectively, and $V_1$ and $V_2$ are the Poisson's ratios of samples.

When the vertical load is 13,000 N, the corresponding Hertzian stress was 6 GPa.

## 3. Results

### 3.1. Determining Fatigue Life

CF tests of different running periods were carried out to obtain the surface integrity parameters of the 18CrNiMo7-6 steel samples at different stages. The premise of dividing the fatigue life of the sample reasonably was based on obtaining the fatigue life distribution curve. Six groups of fatigue tests were carried out to obtain the mean life of the samples. The details of the experimental results of fatigue life are shown in Table 1. The fatigue life distribution curve of the 18CrNiMo7-6 steel samples is shown in Figure 2.

**Table 1.** Fatigue life distribution of 18CrNiMo7-6 steel samples.

| Sample | NO.1 | NO.2 | NO.3 | NO.4 | NO.5 | NO.6 | Mean Life |
|---|---|---|---|---|---|---|---|
| Fatigue life | $8.23 \times 10^6$ | $9.35 \times 10^6$ | $1.35 \times 10^7$ | $2.55 \times 10^7$ | $4.77 \times 10^7$ | $5.04 \times 10^7$ | $3.80 \times 10^7$ |

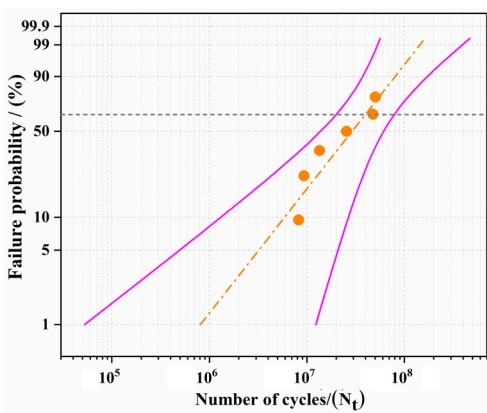

**Figure 2.** Fatigue life distribution curve of the 18CrNiMo7-6 steel.

As can be seen from Table 1 and Figure 2, the mean life of the 18CrNiMo7-6 steel samples was $3.80 \times 10^7$ cycles, and most samples have fatigue failure of more than $10^7$ cycles in the tests. Therefore, the study on the evolution law of the surface integrity of the samples before fatigue failure was of great significance for revealing the fatigue failure mechanisms and improving the fatigue life. Based on the fatigue life obtained, the tested samples were divided into different numbers of running cycles, namely the initial state, 0 cycles (Sample 1), $3 \times 10^6$ cycles (Sample 2), $5 \times 10^6$ cycles (Sample 3), $8 \times 10^6$ cycles (Sample 4) and $3.8 \times 10^7$ cycles (Sample 5), to study the surface integrity evolution characteristics.

### 3.2. Microstructure

Figure 3 presents the cross-sectional EBSD microstructure, average grain size and pole figures near the fatigue failure location after different numbers of running cycles. As shown in Figure 3(a1), in the sample with coarse grains in the initial state, the grain size was

distributed in the range of 0~25 µm, and 74.79% of the grains were smaller than 5 µm. The average grain size was 0.94 µm, as presented in Figure 3(a2). In addition, it can be observed that the layered structures were distributed along the vertical loading direction. Figure 3(a3) presents the (001), (011) and (111) pole figures with a maximum intensity value close to 7.17. After operation, the original coarse grains were slightly refined, as shown in Figure 3(a1–e1), and grain size was decreased after contact operation. As shown in Figure 3(a2–e2), the average grain size of the samples with $3 \times 10^6$, $5 \times 10^6$, $8 \times 10^6$ and $3.8 \times 10^7$ cycles was 0.82 µm, 0.78 µm, 0.70 µm and 0.67 µm, respectively. It should be noted that the polar density of the samples decreased, suggesting that the texture orientation of the material properties in the initial state was changed, as shown in Figure 3(a3–e3). Wang et al. [20] discovered the deformation characteristics and texture evolution mechanisms of martensite steel, and they found that the low-angle grain boundaries with misorientation angles help to improve fatigue performance. In addition, the average grain size of the samples decreased with an increase in the number of running cycles. At the same time, it could be seen that the proportion of small grain sizes had increased. Also, it can be inferred that with the increase in the number of running cycles, the contact surface morphology of the samples changed, causing different vibration between contact surfaces, resulting in the differentiation of grain size and polar density. Moreover, the vibration effects can deform the surface layer material and significantly accelerate grain refinement. Based on the Hall–Petch relationship [21], the relationship between grain size and sample strength can be obtained, which also would help improve the fatigue behavior. Based on the above conclusion, it can be determined that the running state of the fatigue samples directly affects the characteristic change in the microstructure of the material surface layer, which has important guiding significance for evaluating the service performance of the fatigue samples by using this evolutionary characteristic.

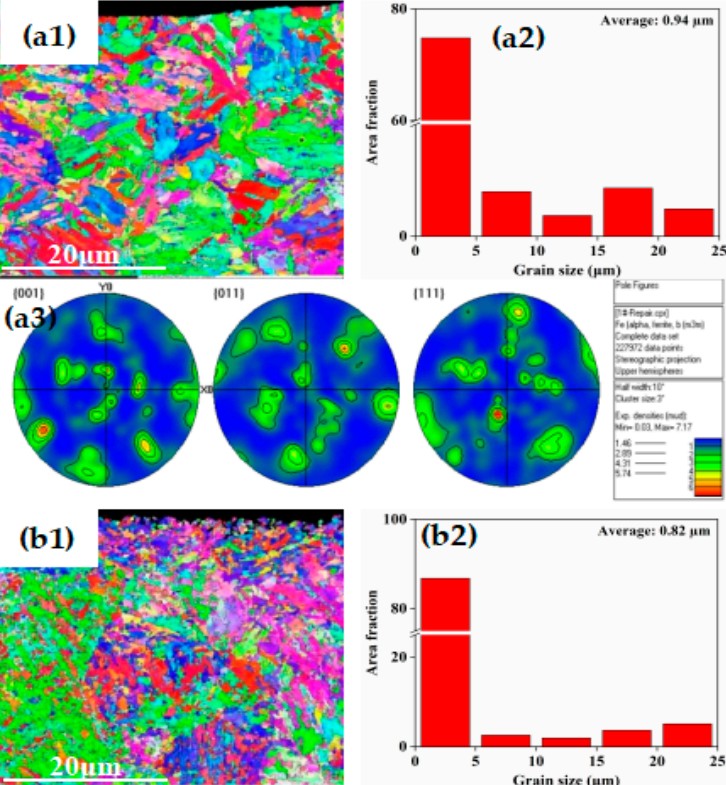

**Figure 3.** *Cont.*

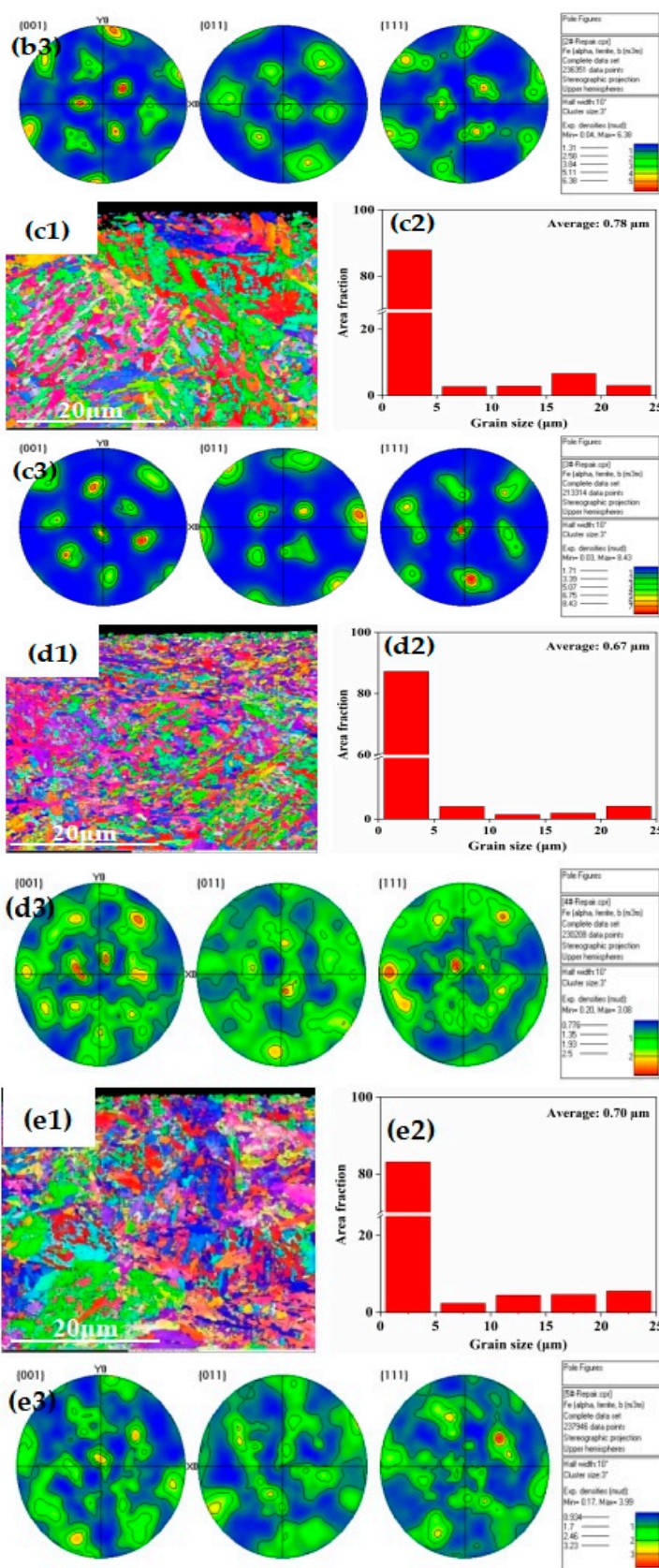

**Figure 3.** EBSD microstructure, average grain size and pole figures of the samples with different numbers of running cycles: (**a1–a3**) Sample 1; (**b1–b3**) Sample 2; (**c1–c3**) Sample 3; (**d1–d3**) Sample 4; (**e1–e3**) Sample 5.

### 3.3. Microhardness and Residual Stress

Figure 4a shows the surface microhardness of the samples with different numbers of running cycles. The initial microhardness of the sample was 696 $HV_{0.2}$, and the microhardness of the samples with $3 \times 10^6$, $5 \times 10^6$, $8 \times 10^6$ and $3.8 \times 10^7$ cycles was 712 $HV_{0.2}$, 720 $HV_{0.2}$, 708 $HV_{0.2}$ and 688 $HV_{0.2}$, respectively. It can be seen that with the increase in the number of running cycles, the surface microhardness of samples increased first and then decreased. The main reasons for the increase in surface microhardness was work hardening, but with the increase in the number of running cycles, the temperature of the contact interface has a softening effect. Cvetkovski et al. [22] studied the thermal softening of fine pearlitic steel and its influence on fatigue properties, and their research conclusions proved that material softening can lead to a decrease in material deformation resistance, resulting in a decrease in fatigue life. Figure 4b shows residual stress at the top surface of the samples with different numbers of running cycles. The surface residual stress in the initial state of the sample was compressed, which was −24 MPa, and the residual stress of the samples with $3 \times 10^6$, $5 \times 10^6$, $8 \times 10^6$ and $3.8 \times 10^7$ cycles was −185 MPa, −277 MPa, −307 MPa and −270 MPa, respectively. The surface residual stress of the samples with different numbers of running cycles was higher than that in the initial state. It should be noted that the residual stress value of Sample 5 was lower than that of Sample 4 and Sample 3, which may be due to the softening effect of the contact surface accelerating the decline in residual stress and material strength with the increase in the number of cycles [23]. This inference was consistent with the conclusion of microhardness analysis.

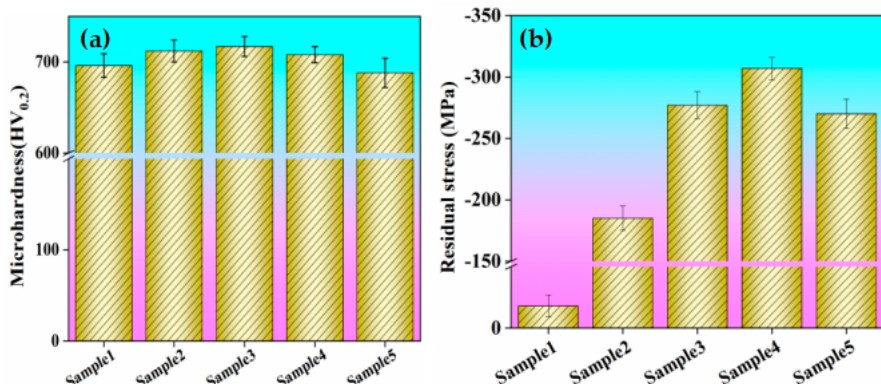

**Figure 4.** (**a**) Surface microhardness of the samples with different numbers of running cycles; (**b**) Surface residual stress of different samples with different numbers of running cycles.

### 3.4. Surface Roughness and Morphology

Figure 5a presents the surface roughness of the samples with different numbers of running cycles, and it can be found that the the surface roughness Ra value has great differences after different numbers of running cycles. The surface roughness Ra value increased first and then decreased with the increase in the number of fatigue cycles. The initial surface roughness value Ra of the Sample 1 was 0.491 μm. The surface roughness of Sample 2, Sample 3, Sample 4 and Sample 5 cycles was 0.206, 0.255, 1.688 and 2.099 μm, respectively. In addition, the surface roughness curve of Sample 1 shows more peaks and troughs, and has high curve fluctuations. With the contact operation of the fatigue samples, the peaks were gradually flattened, and the roughness curves were relatively smooth. Figure 5b presents the surface morphology of the samples with different numbers of running cycles. The machining marks produced in the process of sample preparation were regularly distributed on the sample surface, and this was the reason for the large fluctuation in the surface roughness curve of the original sample. With the increase in the number of running cycles, the machining marks on the surface gradually disappeared, and new surface damage can be observed in Figure 6b,c. Also, local smooth areas were formed, and surface damage started from slight scratches and gradually evolved into pitting until the final spalling. This

was because the machining marks were smoothed out as the number of running cycles increased, while surface damage was formed, resulting in an increase in roughness value. Zhang et al. [24] demonstrated that the surface morphology characteristics are important factors affecting CF performance. It should be noted that the pitting and spalling on the surface of Sample 5 would further expand into delamination failure, resulting in extensive material peeling. According to EDS analysis of the composition distribution of spalling in Figure 6e, no oxygen element was detected, indicating that there was no obvious oxidative wear and fatigue damage on the contact surface before 18CrNiMo7-6 steel severe fatigue failure. Related studies suggested that oxidation is one of the important factors affecting fatigue life [11,25].

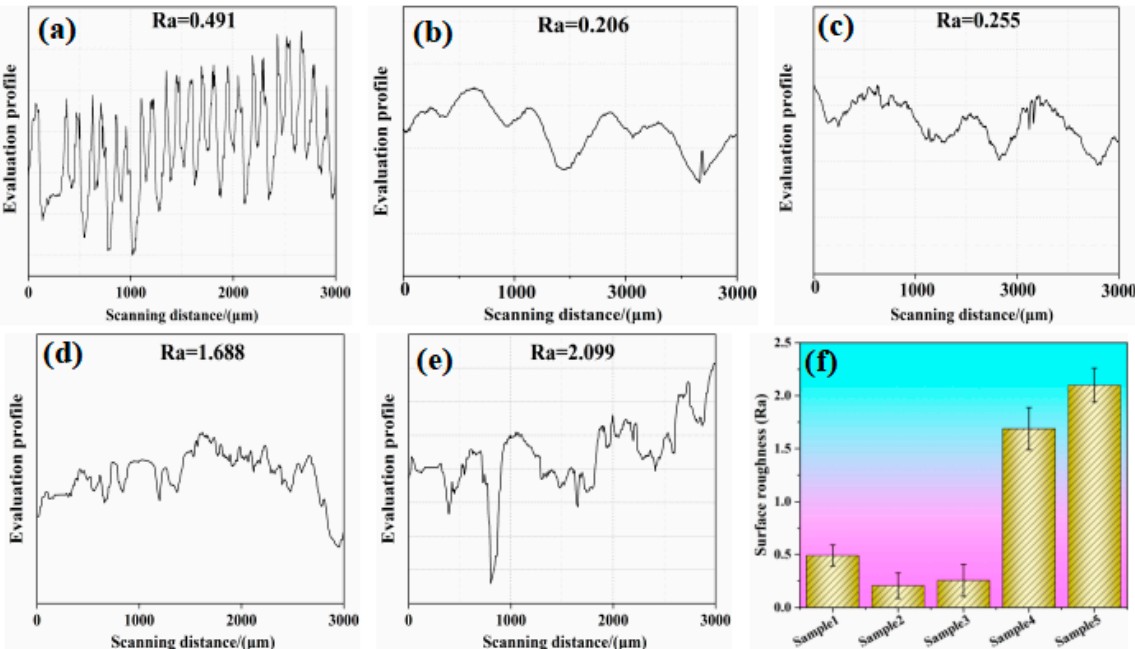

**Figure 5.** Surface roughness of the samples with different numbers of running cycles: (**a**) Sample 1; (**b**) Sample 2; (**c**) Sample 3; (**d**) Sample 4; (**e**) Sample 5; (**f**) surface roughness summary map.

*3.5. Analysis of the Relationship between Surface Integrity and Fatigue Life*

Figure 7 displays the fatigue life evolution relationships between surface roughness, microhardness, average grain size and residual stress. Both surface roughness and microhardness can be obtained using portable equipment, and thus the service performance of the samples can be inferred. The average grain size obtained via EBSD was relatively complex, which was a supplement to the evaluation and fatigue life evolution method. Figure 7a displays the relationships between surface roughness, microhardness and fatigue life. It can be seen that the fatigue life of samples can be clearly evaluated under the coupling relationships of microhardness and surface roughness. In addition, compared with surface roughness, the influence of microhardness on CF life was relatively weak, and when the surface roughness increased from 0.4 μm to 2.0 μm, it gradually approached the fatigue failure point. Moreover, when the surface roughness was greater than 1.6 μm, the influence of microhardness on fatigue life decreased markedly. Figure 7b displays the relationships between average grain size, residual stress and fatigue life. It can be seen that the average grain size was between 0.7 μm and 0.8 μm, and the samples were in a healthy running state. With the accumulation of surface damage, samples produced more severe vibration, and formed an impact effect, which resulted in the further reduction in surface grain size and a greater release of residual stress [26]. Therefore, samples were also close to the fatigue failure point.

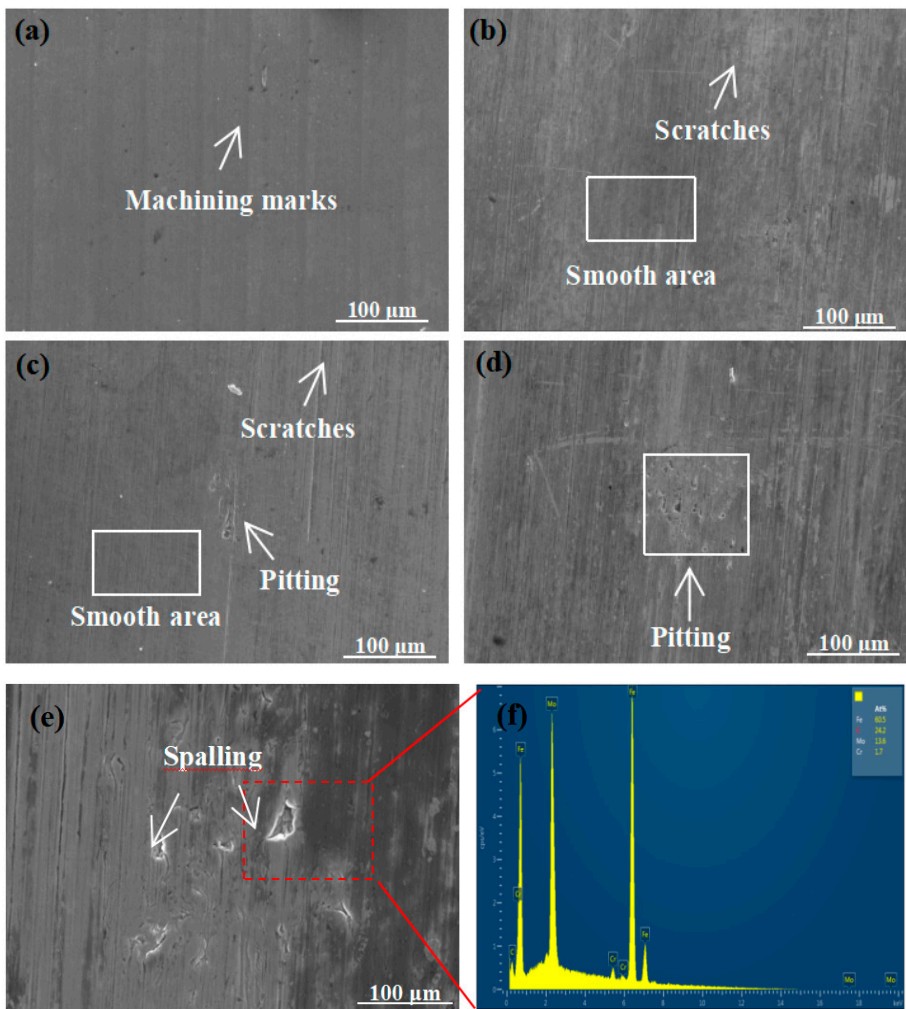

**Figure 6.** Surface morphology and roughness of the samples with different numbers of running cycles: (**a**) Sample 1; (**b**) Sample 2; (**c**) Sample 3; (**d**) Sample 4; (**e**) Sample 5; (**f**) EDS analysis of the Sample 5.

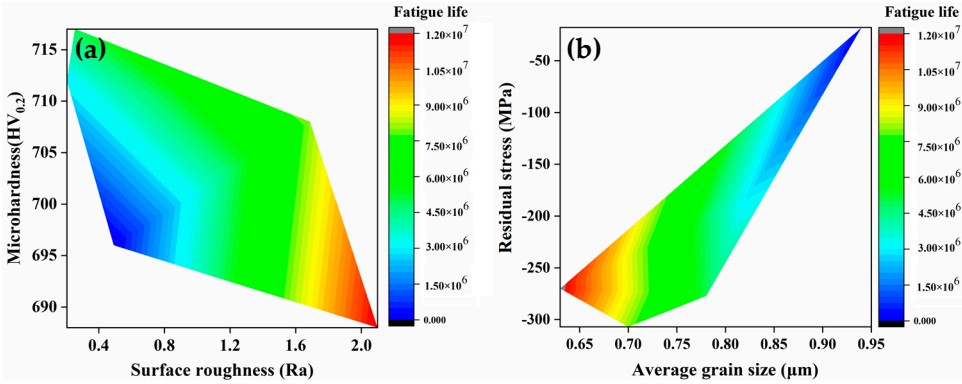

**Figure 7.** The relationships between surface integrity parameters and fatigue limit. (**a**) the relationships between surface roughness, microhardness and fatigue life. (**b**) the relationships between average grain size, residual stress and fatigue life.

## 4. Discussion

By obtaining the surface integrity parameters of 18CrNiMo7-6 gear steel samples with different numbers of running cycles, the variation rules of surface integrity can be identified, providing the most basic data for the prediction of the fatigue life of samples

and the maintenance of a healthy running state. There have been few reports on the use of this method to predict fatigue life and to monitor the operational health status of gears. Once the method is successfully applied, it will provide a new solution for the health monitoring of the core moving components of high-end equipment such as gears and bearings. Future work will further study the effect of different heat treatment processes and running conditions on the surface integrity of 18CrNiMo7-6 gear steel and improve the basic data of the fatigue properties of gear materials.

## 5. Conclusions

In this paper, the surface integrity (surface morphology, microstructure, microhardness, residual stress) characteristics of CF samples based on different numbers of running cycles was investigated. Also, the relationship between surface integrity and fatigue life was analyzed. The main findings are as follows:

(1) A large difference in the surface integrity characteristics of different numbers of running cycles was observed. Surface machining marks were first gradually polished (about $5 \times 10^6$ cycles) and then surface pitting damage was formed (about $8 \times 10^6$ cycles) during the fatigue test.

(2) The average grain size decreased with the increase in the number of running cycles. Within the testing range, the grain size gradually decreased by 0.94 μm from 0.67 μm.

(3) As the number of running cycles increased, the surface microhardness, residual stress and surface roughness Ra increased first and then decreased.

(4) Based on the evolution law of surface integrity mentioned above, the relationships between different surface integrity parameters and fatigue life were plotted. When the surface roughness Ra increased from 0.4 μm to 2.0 μm, it gradually approached the failure point, and the average grain size was between 0.7 μm and 0.8 μm, meaning that the samples were in a healthy running state.

**Author Contributions:** Conceptualization, L.W. and Y.Z.; methodology, Y.L.; data curation, L.W.; writing—original draft preparation, Y.Z.; writing—review and editing, V.J. and A.L.; project administration, L.W.; funding acquisition, Y.Z. All authors have read and agreed to the published version of the manuscript.

**Funding:** This work was supported by the China Postdoctoral Science Foundation (2022M712920), Youth Research Funds Plan of Zhengzhou University of Aeronautics (23HQN01001), Key Scientific Research Project of Colleges and Universities in Henan Province (23A460018) and Henan Province Science and Technology Research Project (232102221021).

**Data Availability Statement:** Data will be made available upon request.

**Acknowledgments:** The authors would like to thank ZKKF (Beijing) Science and Technology Company for supporting the evaluation of microstructure.

**Conflicts of Interest:** The authors declare no conflict of interest.

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
