# Peer review of "Contact Fatigue Behavior Evolution of 18CrNiMo7-6 Gear Steel Based on Surface Integrity"

_metals, doi:10.3390/met13091605_

Round 1

Reviewer 1 Report

Some of my observations, suggestions and corrections are as follows:

General on the topic and data presented in this paper:

Experimental data and findings provided in this research are definitely interesting and could be a good contribution to the research field, i.e. the development of better and more accurate methods for assessing fatigue of gears. It is a shame that bigger number of samples is not used, as it would be better from the statistical point of view. Nevertheless, even with this very limited number of samples, some insight in material processes happening in machine elements during operation is given. This is especially important in cases when there is limited experimental data, even if parts are for the most part durable enough in terms of design requirements.

However, some changes and corrections of errors, and a more clear writing needs to be made before this paper is even considered to be published. Some of my remarks are given below:

General formatting:

Line 14 – seems that “The contact fatigue” is written in bigger font. Correct it.

Line 15 (and all other places where it is mentioned in tekst) – it is not clear to me what exactly is meant with “running cycles”? Is it testing sample surfaces at different cycle count or are these different samples exposed to different cycle count? It should be more clear to the reader.

Line 18 – after “plotted”, correct comma(,) into dot(.).

Line 327 – Seems like this “Disclaimer/” is to be deleted?

Figures and tables:

   There is no Table 4, and it is referenced in line 96? I think this should be Table 1?

   Figure 3 is too large and spans across several pages so I would suggest to the authors to break it up in several pictures (maybe group in one figure all “a” figure (a1, a2, a3, a4), in another all “b” figures and so on. Also, there are some markings missing/misplaced in this figure as is, namely (a2) and (e2). Also, marking in the text (ie. A1) should be the same as in the figure (ie. a1), as it is not currently so.

 Figures should not span to other page, so adjust other pictures as needed (currently there is that problem with Figure 5 also, alongside with some markings staying on page 7 while they should be with part of the figure on page 8)

English language:

Language is for the most part understandable, but some care should be taken to write better and more clearly in English, as there are some sentences that are not clearly and/or correctly written and there are many grammatical and writing errors. For example, just to point out a some of these problems:

Line 32-33 – sentence “The effective …” is not written correctly so I am not sure what the authors mean by this sentence? Sentence should be rewritten clearly.

Line 48, 49 – rephrase to something like “..due to forming of dislocation tangles.”

Line 84 – “Vertical” should not be written in capital letter, and sentence is not correctly written

Line 86 – title should be “Determining Fatigue Life” or something like that.

Line 90,91 – this sentence should be more clearly written, maybe break it up in two sentences.

Line 109-110 – rephrase the sentence

Line 115 – rewrite to “It should also be noted…”

Line 120 – should it be written “…angles are helpful to…”?

Line 150,152 – correct to “increased” (remove “d”)

Line 174 – What is meant with “surface roughness of… …cycles”? Not sure…

Line 174-175 – shouldn’t this sentence be the other way around?

Line 197, 197 – End of this sentence is unclear? What does this mean?

Etc. (I do not with to do all the other grammatical corrections, it is the responsibility of authors of this paper)

Article “the” is inappropriately extensively used. Some (but not all) suggestions for removal of the aforementioned article is in lines 14, 25, 29, 32, 34, 62, 65, 83, 87, 121 etc. Please correct this, it is almost unreadable this way.

Also, a lot of inappropriate use of “And” as a start of a new sentence… just to give attention to few, it can be found in lines 13, 74, 91 etc.

There is also a lot of mixup with “was”/”were” (singular/plural) through the whole text, problems with damage/damages and similar. It should all be corrected.

Author Response

Dear Reviewer

Thanks for reviewing our manuscript (metals-2546711). We have carefully modified the manuscript and the main changes were marked in yellow background in the revised manuscript.

Experimental data and findings provided in this research are definitely interesting and could be a good contribution to the research field, i.e. the development of better and more accurate methods for assessing fatigue of gears. It is a shame that bigger number of samples is not used, as it would be better from the statistical point of view. Nevertheless, even with this very limited number of samples, some insight in material processes happening in machine elements during operation is given. This is especially important in cases when there is limited experimental data, even if parts are for the most part durable enough in terms of design requirements.

However, some changes and corrections of errors, and a more clear writing needs to be made before this paper is even considered to be published. Some of my remarks are given below:

General formatting:

Line 14 – seems that “The contact fatigue” is written in bigger font. Correct it.

Response to Comment:

Thanks for your comments. As Reviewer suggested, we have modified “ contact fatigue” to “ CF”

Line 15 (and all other places where it is mentioned in tekst) – it is not clear to me what exactly is meant with “running cycles”? Is it testing sample surfaces at different cycle count or are these different samples exposed to different cycle count? It should be more clear to the reader.

Response to Comment:

The meaning of “running cycles” is testing sample surfaces at different cycle count. For clarity, we have added the following in the Materials and Methods, As follows:

Therefore, the aim of this work is attempts to analyze the variation of surface integrity in different running cycles, which meant the testing samples surfaces at different cycle count. And the results would provide basic data for providing tooth surface characteristic parameters and later gear maintenance.

Line 18 – after “plotted”, correct comma(,) into dot(.).

Response to Comment:

Thanks for your comments. As Reviewer suggested, we have revised it.

Line 327 – Seems like this “Disclaimer/” is to be deleted?

Response to Comment:

Thanks for your comments. As Reviewer suggested, we have deleted “Disclaimer/”.

Figures and tables:

–    There is no Table 4, and it is referenced in line 96? I think this should be Table 1?

Response to Comment:

Thanks for your comments. It is Table 1, and we have made modifications in the manuscript.

–   Figure 3 is too large and spans across several pages so I would suggest to the authors to break it up in several pictures (maybe group in one figure all “a” figure (a1, a2, a3, a4), in another all “b” figures and so on. Also, there are some markings missing/misplaced in this figure as is, namely (a2) and (e2). Also, marking in the text (ie. A1) should be the same as in the figure (ie. a1), as it is not currently so.

Response to Comment:

Thanks for your comments. The number of pages were reduced by reducing the Figure 3 image size. And the a2 and e2 have been added.

–  Figures should not span to other page, so adjust other pictures as needed (currently there is that problem with Figure 5 also, alongside with some markings staying on page 7 while they should be with part of the figure on page 8)

Response to Comment:

Thanks for your comments. We have adjusted the figures size to ensure that it was within the same page number.

English language:

Language is for the most part understandable, but some care should be taken to write better and more clearly in English, as there are some sentences that are not clearly and/or correctly written and there are many grammatical and writing errors. For example, just to point out a some of these problems:

Line 32-33 – sentence “The effective …” is not written correctly so I am not sure what the authors mean by this sentence? Sentence should be rewritten clearly.

Response to Comment:

Thanks for your comments. The meaning of this sentence is that accurate gear surface integrity feature data can provide effective maintenance and improve the service life of gears. To avoid ambiguity, we have deleted this sentence.

Line 48, 49 – rephrase to something like “..due to forming of dislocation tangles.”

Response to Comment:

Thanks for your comments. As Reviewer suggested, we have revised it.

Line 84 – “Vertical” should not be written in capital letter, and sentence is not correctly written

Response to Comment:

Thanks for your comments. As Reviewer suggested, we have revised it. And the sentence have made modifications in the manuscript. As follows:

When the vertical load is 13000 N, the corresponding Hertzian stress was 6 GPa.

Line 86 – title should be “Determining Fatigue Life” or something like that.

Response to Comment:

Thanks for your comments. As Reviewer suggested, we have revised it.

Line 90,91 – this sentence should be more clearly written, maybe break it up in two sentences.

Response to Comment:

Thanks for your comments. The sentence was divided into two sections.

Line 109-110 – rephrase the sentence

Response to Comment:

Thanks for your comments. As Reviewer suggested, we have revised it. As follows:

In addition, it can be observed that the layered structures were distributed along the vertical loading direction.

Line 115 – rewrite to “It should also be noted…”

Response to Comment:

As Reviewer suggested, we have revised it.

Line 120 – should it be written “…angles are helpful to…”?

Response to Comment:

As Reviewer suggested, we have revised it. As follows:

The low angle grains boundaries with misorientation angles are help to improve fatigue performance.

Line 150,152 – correct to “increased” (remove “d”)

Response to Comment:

As Reviewer suggested, we have removed it.

Line 174 – What is meant with “surface roughness of… …cycles”? Not sure…

Response to Comment:

Thanks for your comments. We have removed it. As follows:

The surface roughness of Sample2, Sample3, Sample4 and Sample5 cycles were 0.206, 0.255, 1.688 and 2.099 μm, respectively.

Line 174-175 – shouldn’t this sentence be the other way around?

Response to Comment:

As Reviewer suggested, we have revised the sentence. As follows:

The surface roughness Ra value increased first and then decreased with the increased of the number of fatigue cycles. The initial surface roughness value Ra of the Sample1 was 0.491 μm. The surface roughness of Sample2, Sample3, Sample4 and Sample5 cycles were 0.206, 0.255, 1.688 and 2.099 μm, respectively.

Line 197, 197 – End of this sentence is unclear? What does this mean?

Response to Comment:

We have revised the sentence. As follows:

Related studies suggested that oxidation was one of the important factors affecting fatigue life.

Etc. (I do not with to do all the other grammatical corrections, it is the responsibility of authors of this paper)

Article “the” is inappropriately extensively used. Some (but not all) suggestions for removal of the aforementioned article is in lines 14, 25, 29, 32, 34, 62, 65, 83, 87, 121 etc. Please correct this, it is almost unreadable this way.

Also, a lot of inappropriate use of “And” as a start of a new sentence… just to give attention to few, it can be found in lines 13, 74, 91 etc.

There is also a lot of mixup with “was”/”were” (singular/plural) through the whole text, problems with damage/damages and similar. It should all be corrected.

Response to Comment:

Thanks for your comments. As Reviewer suggested, we have revised all the comments, checked the whole manuscript, and corrected the mistakes of  the “was”/”were” and “the” .

Reviewer 2 Report

The paper considers surface integrity in contact fatigue samples subjected to varying running cycles with a focus on surface morphology, microstructure, microhardness, and residual stress. The objective was to develop a fatigue life evaluation method based on the evolution of surface integrity under typical working conditions. Main results include analysis of surface integrity variation, grain size reduction with an increase in the number of running cycles, surface microhardness, residual stress and surface roughness change, relationships between different surface integrity parameters and fatigue life.

Research of rolling fatigue and mechano-rolling fatigue under simultaneous action of contact and non contact forces that included the creation of fatigue curves, microhardness and roughness analysis should be referred to in the paper:

Doelling KL, Ling FF, Bryant MD, et al. An Experimental Study of the Correlation between Wear and Entropy Flow in Machinery Components. Journal of Applied Physics 2000; 88. Ling FF, Bryant MD, Doelling KL. On Irreversible Thermodynamics for Wear Prediction. Wear 2002; 253: 1165–1172. Beheshti A, Khonsari MM. A Thermodynamic Approach for Prediction of Wear Coefficient Under Unlubricated Sliding Condition. Tribology Letters 2010; 38: 347–354.  Amiri M, Khonsari MM. On the Thermodynamics of Friction and Wear – A Review. Entropy 2010; 12: 1021–1049. Beheshti A, Khonsari MM. On the Prediction of Fatigue Crack Initiation in Rolling/Sliding Contacts with Provision for Loading Sequence Effect. Tribology International 2011; 44: 1620–1628.

I suggest to rename “Hertzian stress” in (1) since it is in essence the maximum contact pressure and should be denoted by p0.

What was the limiting state (failure criterion) for rolling fatigue life estimation (fig. 2) of the roller/roller system? Was it the appearance of the first pitting and if so what was its size?

The paper “Contact Fatigue Behavior Evolution of 18CrNiMo7-6 Gear Steel Based on Surface Integrity” could be considered for publication in Metals only after addressing the above comments.

No significant issues.

Author Response

Dear Reviewer

Thanks for reviewing our manuscript (metals-2546711). We have carefully modified the manuscript and the main changes were marked in green background in the revised manuscript.

The paper considers surface integrity in contact fatigue samples subjected to varying running cycles with a focus on surface morphology, microstructure, microhardness, and residual stress. The objective was to develop a fatigue life evaluation method based on the evolution of surface integrity under typical working conditions. Main results include analysis of surface integrity variation, grain size reduction with an increase in the number of running cycles, surface microhardness, residual stress and surface roughness change, relationships between different surface integrity parameters and fatigue life.

Research of rolling fatigue and mechano-rolling fatigue under simultaneous action of contact and non contact forces that included the creation of fatigue curves, microhardness and roughness analysis should be referred to in the paper:

Doelling KL, Ling FF, Bryant MD, et al. An Experimental Study of the Correlation between Wear and Entropy Flow in Machinery Components. Journal of Applied Physics 2000; 88. Ling FF, Bryant MD, Doelling KL. On Irreversible Thermodynamics for Wear Prediction. Wear 2002; 253: 1165–1172. Beheshti A, Khonsari MM. A Thermodynamic Approach for Prediction of Wear Coefficient Under Unlubricated Sliding Condition. Tribology Letters 2010; 38: 347–354.  Amiri M, Khonsari MM. On the Thermodynamics of Friction and Wear – A Review. Entropy 2010; 12: 1021–1049. Beheshti A, Khonsari MM. On the Prediction of Fatigue Crack Initiation in Rolling/Sliding Contacts with Provision for Loading Sequence Effect. Tribology International 2011; 44: 1620–1628.

Response to Comment:

Thanks for your comments. Considering the relevance of the research and the publication time of the paper, “On Irreversible Thermodynamics for Wear Prediction” and “On the Prediction of Fatigue Crack Initiation in Rolling/Sliding Contacts with Provision for Loading Sequence Effect” have been cited. (Ref 17 and Ref 18).

I suggest to rename “Hertzian stress” in (1) since it is in essence the maximum contact pressure and should be denoted by p0.

Response to Comment:

Thanks for your comments. As Reviewer suggested, we have revised it. As follows:

What was the limiting state (failure criterion) for rolling fatigue life estimation (fig. 2) of the roller/roller system? Was it the appearance of the first pitting and if so what was its size?

Response to Comment:

Thanks for your comments. As described in the “Materials and Methods”, fatigue failure depends on the system detecting vibration signals value. When fatigue failure ( pitting/ Spalling) first occurred on the contact surface, the vibration signals value increased, and the fatigue testing machine would be automatically stop and save the data. But, due to the anisotropic characteristics of 18CrNiMo7-6 gear steel, fatigue failure does not have a fixed size.

Reviewer 3 Report

Authors deal with the problem of contact fatigue of a gear, using surface integrity as an indicator. The study seems reasonably well designed, and the conclusions are backed by obtained results. However, some minor improvements might be beneficial.

1. Introduction section can be expanded with significant cases of gear failures and lessons learned.

2. Addition of macro post-testing images of gears would be appreciated.

3. Discussion and Conclusion sections could be joined and discussion further improved with comparing the obtained results with same/similar ones of other authors.

4. A note about possible use of the results in numerical modelling of fatigue could be usefu.

5. References should include publications beyond regional interest.

Moderate corrections needed.

Author Response

Dear Reviewer

Thanks for reviewing our manuscript (metals-2546711). We have carefully modified the manuscript and the main changes were marked in red background in the revised manuscript.

Authors deal with the problem of contact fatigue of a gear, using surface integrity as an indicator. The study seems reasonably well designed, and the conclusions are backed by obtained results. However, some minor improvements might be beneficial.

  1. Introduction section can be expanded with significant cases of gear failures and lessons learned.

Response to Comment:

Thanks for your comments. As Reviewer suggested, we have added the cases of gear failures. As follows:

For example, Bejger et al. believed that the gears were subjected to alternating loads due to wind speed changes and free braking pulses, which makes them one of the most fragile components of a wind turbine with low reliability, and the gears fatigue failure led to abnormal operation of wind power generation devices[6]. 

  1. Addition of macro post-testing images of gears would be appreciated.

Response to Comment:

Thanks for your comments. At present, our main research is to carry out research on gear materials. After verifying a large number of data, we will further study gear samples in order to save costs

  1. Discussion and Conclusion sections could be joined and discussion further improved with comparing the obtained results with same/similar ones of other authors.

Response to Comment:

Thanks for your comments. As Reviewer suggested, the comparison results should be added to improved the quality of the Discussion and Conclusion sections. The results with same/similar ones of other authors have been explained in the introduction section. As follows:

Although a series of research achievements have been made in improving the surface integrity, fatigue life and service performance of gear samples, the variation law of tooth surface integrity during service period has not yet been known. Therefore, the aim of this work is attempts to analyze the variation of surface integrity in different running cycles, which meant the testing samples surfaces at different cycle count. And the results would provide basic data for providing tooth surface characteristic parameters and later gear maintenance.

  1. A note about possible use of the results in numerical modelling of fatigue could be useful.

Response to Comment:

Thanks for your comments. The test data is an important basis for the construction of failure numerical model. That's what we're going to do next.

  1. References should include publications beyond regional interest.

Response to Comment:

Thanks for your comments. We added to the References on cracks initiation and fatigue life prediction based on surface damages and wear performance [17,18], which were publications beyond regional interest.

Round 2

Reviewer 1 Report

Experimental data and findings provided in this research are definitely interesting and could be a good contribution to the research field. In this form/writing article can be easily undestood and it can be considered for publishing after minor/moderate language corrections.

Of non-language problems, I just noticed the following:

Line 228 (Figure 7) – move (a) and (b) markings as to not overlap with graphic of pictures.

Some of my language corrections in a second round of review are as follows:

Line 123-124 – Rephrase:  “…and they found that the low angle grains boundaries with misorientation angles are help to improve fatigue performance. “ to something like this: “..and they found that the low angle grain boundaries with misorientation angles help to improve fatigue performance.”

Line 125 – change “increased “ to “increase”

Line 125 – Rephrase “And the proportion of small grain size was increased.“  to something like “At the same time it could be seen that  the proportion of small grain size has increased. “

Line 126 - correct increased to increase (remove “d”) in  “the increased of running cycles”

Line 155 – correct “work hardening, but with the increased of…“ to „increase“

Line 177, 186, 249 – correct “increased” to “increase”

Line 187, 188, 191, 197, 224, 248 – correct “damages” to “damage”

Line 235 – change “predicte” to “predict”

Line 237, 238 – change “And the next work would further…” to “Future work will further…”

Line 248 – change “were” to “was”

Author Response

Dear Reviewer

Thanks for reviewing our manuscript (metals-2546711). We have carefully modified the manuscript and the main changes were marked in yellow background in the revised manuscript.

Experimental data and findings provided in this research are definitely interesting and could be a good contribution to the research field. In this form/writing article can be easily undestood and it can be considered for publishing after minor/moderate language corrections.

Of non-language problems, I just noticed the following:

Line 228 (Figure 7) – move (a) and (b) markings as to not overlap with graphic of pictures.

Response to Comment:

Thanks for your comments. As Reviewer suggested, we have revised it in the manuscript.

Comments on the Quality of English Language

Some of my language corrections in a second round of review are as follows:

Line 123-124 – Rephrase:  “…and they found that the low angle grains boundaries with misorientation angles are help to improve fatigue performance. “ to something like this: “..and they found that the low angle grain boundaries with misorientation angles help to improve fatigue performance.”

Response to Comment:

Thanks for your comments. As Reviewer suggested, we have revised it. As follows:

Wang et al. [20] discovered the deformation characteristics and texture evolution mechanisms of martensite steel, and they found that the low angle grain boundaries with misorientation angles help to improve fatigue performance.

Line 125 – change “increased “ to “increase”

Thanks for your comments. As Reviewer suggested, we have revised it.

Line 125 – Rephrase “And the proportion of small grain size was increased.“  to something like “At the same time

it could be seen that the proportion of small grain size has increased. “

Thanks for your comments. As Reviewer suggested, we have revised it. As follows:

At the same time, it could be seen that  the proportion of small grain size has increased.

Line 126 - correct increased to increase (remove “d”) in  “the increased of running cycles”

Thanks for your comments. As Reviewer suggested, we have revised it.

Line 155 – correct “work hardening, but with the increased of…“ to „increase“

Thanks for your comments. As Reviewer suggested, we have revised it.

Line 177, 186, 249 – correct “increased” to “increase”

As Reviewer suggested, we have revised it.

Line 187, 188, 191, 197, 224, 248 – correct “damages” to “damage”

Response to Comment:

Thanks for your comments. As Reviewer suggested, we have revised it.

Line 235 – change “predicte” to “predict”

Response to Comment:

Thanks for your comments. As Reviewer suggested, we have revised it.

Line 237, 238 – change “And the next work would further…” to “Future work will further…”

Response to Comment:

Thanks for your comments. As Reviewer suggested, we have revised it. As follows:

Future work will further study the effect of different heat treatment processes and running conditions on the surface integrity of 18CrNiMo7-6 gear steel, and improve the basic data of fatigue properties of gear materials.

Line 248 – change “were” to “was”

As Reviewer suggested, we have revised it.

Reviewer 2 Report

.

Author Response

Thank the reviewers and editors for their comments on the manuscript, Reviewer 2 gave a positive answer to each comment. And the quality of the manuscript has been improved.